# Effects of Sound Interventions on the Permeability of the Blood–Brain Barrier and Meningeal Lymphatic Clearance

**DOI:** 10.3390/brainsci12060742

**Published:** 2022-06-05

**Authors:** Sean Sachdeva, Sushmita Persaud, Milani Patel, Peyton Popard, Aaron Colverson, Sylvain Doré

**Affiliations:** 1Department of Anesthesiology, Center for Translational Research in Neurodegenerative Disease, College of Medicine, University of Florida, Gainesville, FL 32610, USA; seansachdeva@ufl.edu (S.S.); s.persaud@ufl.edu (S.P.); milanipatel@ufl.edu (M.P.); peytonpopard@ufl.edu (P.P.); 2Musicology/Ethnomusicology Program, School of Music, College of the Arts, University of Florida, Gainesville, FL 32603, USA; acolverson@ufl.edu; 3Departments of Pharmaceutics, Psychology, and Neuroscience, McKnight Brain Institute, College of Medicine, University of Florida, Gainesville, FL 32610, USA

**Keywords:** elimination, glymphatic system, meninges, music, noise, opening

## Abstract

The meningeal lymphatic, or glymphatic, system is receiving increasing attention from the scientific community. Recent work includes noninvasive techniques to demonstrate relationships between blood–brain barrier (BBB) activity and the glymphatic system in the human central nervous system. One potential technique is the use of music/sound to enhance BBB permeability regarding the movement of small molecules in and out of the brain. However, there is minimal knowledge regarding the methodical investigation(s) of the uses of music/sound on BBB permeability and glymphatic clearance and the outcomes of these investigation(s). This review contains evidence discussing relationships between music/sound, BBB permeability, and meningeal lymphatic clearance. An overview of the anatomy and physiology of the system is presented. We discuss the uses of music/sound to modulate brain and body functions, highlighting music’s effects on mood and autonomic, cognitive, and neuronal function. We also propose implications for follow-up work. The results showed that music and sound interventions do, in fact, contribute to the opening of the BBB and subsequently increase the function of the meningeal lymphatic system. Evidence also suggests that music/sound has the ability to reduce the collateral effects of brain injuries. Unfortunately, music/sound is rarely used in the clinical setting as a medical intervention. Still, recent research shows the potential positive impacts that music/sound could have on various organ systems.

## 1. Introduction

The glymphatic system was discovered relatively recently; however, the existence of the lymphatic system/vessels has been known for centuries. Re-discoveries of different portions of the system since the 1950s have led researchers to identify the overall role of the lymphatic pathway. This pathway serves as a connection between the interstitial space and the peripheral tissues; it was soon determined that this connection allowed molecules to drain from cells of vital organs throughout the body [1,2]. In the past decade, researchers characterized a waste clearance system that relies on perivascular channels sourced from astroglial cells in the brain, allowing metabolites and soluble proteins to be eliminated from the central nervous system [3]. This pathway was categorized as the glymphatic system because of its association with the brain’s glial cells and brain meninges. In addition to waste clearance, the glymphatic system diffuses crucial molecules such as glucose, neurotransmitters, lipids, and amino acids throughout the brain [3]. This is visualized in Figure 1, in which the pathway of waste metabolites is shown in terms of the anatomical view of the glymphatic/meningeal lymphatic system (MLS). Of interest, its flow or pulse wave appears to be activated during regular sleep patterns. For example, various theories have been proposed to better understand and accelerate the clearance of these toxic Aβ splicing fragments from the brain [3]. The glymphatic system exists in the perivascular space and is now considered to contribute to approximately 60% of the brain’s Aβ drainage to the cervical lymph nodes and further downstream [4]. However, because of new technologies, this is being rigorously re-evaluated and independently validated; much remains unknown.

### 1.1. Music and Body and Brain Function Health

Music is receiving increasing attention for its beneficial effects on a variety of health outcomes. One area of recent attention is severe mental illness [5]. Golden and colleagues conducted a global scoping review of evidence on the uses of music to treat or mitigate symptoms of severe mental illness. Their results point to the significant benefits of using music to manage symptoms of major mental disorders, with over two-thirds of included studies reporting positive effects. The study, however, concluded that the heterogeneity of the study design limited the replicability and transferability of the findings to various populations of interest, and the authors provided five recommendations for follow-up work [5]. A separate study examined the effects of music on preoperative anxiety and incorporated 26 trials (2051 participants). Music exposure resulted in an average reduction in anxiety by 5.72 units of measure compared to the standard control group [6]. Studies examining the effects of music on anxiety reduction in patients with coronary heart disease have reported similar results [6]. Similar decreases have also been reported in mechanically ventilated patients [7] and cancer patients [8]. Music, in this instance, can function similarly to anxiety-reducing medications, indicating the need for research on how music can affect the brain in terms of nervous system function and response.

### 1.2. Music and Autonomic Function

There is substantial research on the effects of music on autonomic nervous system (ANS) activity. The ANS is the main driving force in regulating homeostasis and adaptation of the body based on external circumstances. Incorporating music can have positive effects that have yet to be explored. Extensive research has shown that music affects physiological variables associated with blood pressure, heart rate, respiratory rate, body temperature, and various biochemical parameters. In addition to these findings, similar effects have been reported for the cardiac ANS. Music stimulus increases parasympathetic activity associated with increased heart rate variability, which is the variation of time intervals between successive heartbeats [9]. Across 29 studies with varying populations and music therapies, all but three groups showed initial results of a positive impact of music on heart rate variability (*p* = 0.05). Thus, data in this specific field are consistent and could form a foundation for researching the effects of music on the glymphatic system.

### 1.3. Music and Cognitive Function

Research shows positive effects of engagement with music and healthy cognitive function. For example, one study showed that music improvisation significantly improved memory [10]. One hundred and thirty-two individuals aged 60 to 90 years old participated in the study, 51 of whom were musicians who had 5 or more years of formal music training. After acquiring neural or emotional information, the experimental and control groups were exposed to music imitation or improvisation. Memory was evaluated based on free recall and recognition using immediate and deferred measurements. The results suggested that a focal musical activity can help enhance memory in older adults. In addition, other studies have shown that listening to music advances cognitive skills such as recognition memory [11], working memory [12], and fluency [13]. Furthermore, another study found an effect of musical training on verbal memory and visual memory in older adults [14]. The study consisted of 24 people, with a median age of 77 years and 3 months. Participants were randomly assigned to music, literature, or untrained groups. The musically trained individuals remembered more words from a list than both control groups, along with symbol sequences. However, this study did not coincide with the influence of music training on working memory.

Previous studies have shown a relationship between music training and central executive processing [15]. Bugos (2019) examined the effects of bimanual coordination in piano performance on cognitive performance in healthy older adults. The study was composed of 135 healthy older adults between the ages of 60 and 80 years who experienced music training interventions. Participants were evaluated using the Wechsler Abbreviated Scale of Intelligence, and their Verbal Intelligence Quotient was measured. In addition, the Advanced Measures of Music Audiation aptitude test was completed to measure musical aptitude. The results suggested that dynamic music performance can benefit working memory, and the extent of the benefit relies on coordination demands. Specifically, the Group Piano Instruction (GPI) and Group Percussion Instruction (GPeI) groups had significantly enhanced performance in visual scanning and working memory compared to the Music Listening Instruction (MLI) group. The GPI group also had an increased processing speed.

### 1.4. Music and Neuronal Function/Health

The existing experimental literature demonstrates that music substantially affects neuronal function and body health. Exposure to learning music enhances specific brain areas, such as the premotor and cerebral cortex, cerebellum, and left Heschl’s gyrus [16]. Heschl’s gyrus is the human primary auditory cortex and an area of acoustic processing. Findings suggest that professional musicians have greater gray matter volume than non-musicians in Heschl’s gyrus [16]. By studying exposure to music in Heschl’s gyrus, new insight may be provided into how music can affect secondary structures. An example is how exposure to music could decrease blood pressure via dynamic interactions between calcium-dependent dopamine synthesis and calmodulin [17]. Music exposure has been shown to positively affect the body by stimulating the development of axonal and dendritic growth and regulating blood pressure.

### 1.5. Relationships between Music, Body/Brain, and Glymphatic Clearance

The relationship between music/sound and glymphatic clearance has seldom been studied because of its novelty. Figure 2 demonstrates the potential overall outcomes of music/sound on the MLS. However, music/sound has the potential effect of permeating the BBB, allowing specific solutes to pass into the brain at a more efficient rate and reach the intended tissue. This passage creates optimal results with a lower dose of medication in a shorter amount of time. Sound-based interventions, specifically ultrasound, have proved successful in increasing the permeability of the blood–brain vasculature. This is because the pores and intracellular gaps within the BBB endothelium are highly dependent on acoustic pressure [18].

An example of how music/sound affects the permeability of the BBB is reported by Semyachkina-Glushkovskaya et al. [19]. Upon intravenously injecting Evans blue (EB) dye into mice, audible sound exposure was applied at 100 dB and 370 Hz. Leakage of the EB through the BBB increased 19.7-fold compared to the group with no exposure to sound (*p* < 0.05). EB is a common macromolecule used to analyze the permeability of the BBB. Because of the dye’s high affinity to serum albumin, essentially all EB is bound to albumin, which does not readily cross the BBB. In a usual permeability scenario, the neural tissue will remain unstained with EB bound to albumin; however, with compromised permeability of the BBB, EB is found in the brain tissue. In terms of fluorescein isothiocyanate (FITC)-dextran, its reaction with primary amines allows it to readily cross the BBB. It is also helpful in analyzing the extravasation of neurological pathways.

Another study exposed mice to Scorpions’ “Still Loving You” at 100 dB [20]. One hour after music exposure, BBB permeability increased 17.3-fold. Importantly, however, this permeability increase was induced only at 90-dB and 100-dB sound levels. The experimental group exposed to music at 70 dB experienced the same rise in BBB permeability as controls.

## 2. Purpose

### Scientific Rationale

The studies mentioned above showed that music affects the function and health of the body and brain, autonomic and cognitive function, and neuronal function and health, often positively. In addition, there is preliminary evidence regarding the effects of music/sound on the activity of the glymphatic system [19,20]. Several studies have also researched the effects of ultrasound on glymphatic clearance [21,22,23]. However, there is limited research in these areas and, therefore, a need to expand such lines of inquiry.

This review aimed to document and discuss the effects of music/sound on BBB permeability and glymphatic clearance. Our research question was: What is the evidence on the benefits of music on BBB permeability and the glymphatic system? This review intends to promote further preclinical and, eventually, clinical research on the connections between music/sound and BBB permeability and glymphatic clearance.

## 3. Methods

### 3.1. Design

This is a narrative review of the current literature on studies regarding music/sound exposure, BBB permeability, and glymphatic clearance. This review was modeled on Wong’s description of constructing a meta-narrative review [24]. This approach “seeks to illuminate a heterogeneous topic area by highlighting the contrasting and complementary ways researchers have studied the same or a similar topic” [24]. Existing literature was summarized and formatted into a table, including results and limitations. In looking at BBB permeability due to music/sound exposure, measurements such as sound levels, sound sources, and the affected tissues were collected. In addition, the review focused on the uses of music/sound to affect brain function and health, autonomic and cognitive function, and neuronal function and health. The themes of this review include sound in health care spaces; sound and stress; cognitive function; music and its effects on the BBB; music as an alternative or supplemental medicine for improving health care; and implications for clinical practice. Other pertinent content regarding shaping an understanding of the field, such as cognitive function and stress, was retrieved, scanned, and added to this review. Common limitations were also included.

### 3.2. Search Strategies

An in-depth literature search was conducted using electronic databases, including Google, Web of Science, Scholar, Embase, Dimensions, and One Search, for relevant articles on studies related to the effects of music on glymphatic clearance, with particular reference to the BBB as it relates to the glymphatic system. Each database was searched for articles published from when the database was created through January 2022. In addition, all available databases were meticulously searched for published articles for all relevant literature.

Various search terms were used to identify similar publications in combination with sound OR noise OR music. The final search terms were music; sound; noise; frequency of music; music therapy; mood; glymphatic system; cerebrospinal fluid; lymphatic; blood–brain barrier; and any additional terms derived from the existing papers. The initial search terms included frequency and ultrasound. Next, the search was narrowed to focus on sound and noise to the glymphatic system. Finally, snowballing of search term strategy development addressed gaps in existing literature, particular to music/sound exposure and the glymphatic system. MeSH terms were created once a link had been established in which the glymphatic system was affecting other parts of the body. The MeSH terms were categorized as follows: ‘Glymphatic System,’ ‘Music,’ ‘Sleep,’ ‘Depression and Glymphatic,’ ‘Neurodegenerative diseases,’ and ‘Ultrasonography.’ Each category had a list of words that were used to find articles.

### 3.3. Inclusion/Exclusion Criteria

Studies that met the following criteria were considered for inclusion: (1) experimental studies performed in rats, mice, or humans; (2) investigation of the effects of music or audible sound on the BBB and glymphatic system; (3) comparison of the effects of music or sound intervention to a control parameter where no music/sound was applied; (4) full-text article was available for download; (5) written in English; (6) published before 10 December 2021. The limitation to the sound intervention applied was that it had to be the music of any kind or audible sound. Therefore, focused ultrasound (FUS) studies were not considered for this review.

## 4. Results

The literature search resulted in eight references in the final review. Upon initial examination of the references, there was a total of 42 original research studies and four literature reviews. The original 46 articles documented research in relation to any form of intervention regarding the glymphatic system, either through the BBB or MLS pathway. Based on the total articles collected, 18 articles studied magnetic resonance-guided FUS and its effects on glymphatic opening and BBB permeability. The four literature reviews consisted of systematic reviews of multiple case studies that sought to draw a conclusion on the impact that music, sound, or noise may have on the glymphatic system. The search parameters were narrowed to focus on music interventions and meningeal lymphatic/glymphatic system relationships, resulting in eight references included in this review. This exclusion was done for the sole purpose of creating a focused review with literature directed only toward music/sound interventions and the MLS/glymphatic pathways. However, the other articles that were excluded provided a framework and background in understanding the MLS/glymphatic process and are described in the discussion. Seven out of the eight references used rodents for their test subjects. Included articles were separated into two categories: humans and rodents. Table 1 describes all included studies on rodents, and Table 2 describes all included studies on humans. This differentiated the findings of the articles to facilitate a comparison of the results in different test subjects.

### 4.1. Music/Sound and the Blood–Brain Barrier

Music and sound interventions contribute to the opening of the blood–brain barrier (BBB) and increase the function of the MLS. Such interventions have resulted in opening the BBB at a specific volume range and frequency intensities recorded between 90 and 110 dB and 370 Hz [19]. Four studies, all of which originated from the same researcher, examined the scope of music/sound on the BBB and MLS in terms of cerebrospinal fluid flow and vessel permeability.

Semyachkina-Glushkovskaya et al. [19] conducted a post-clinical analysis monitoring the meningeal lymphatic vessels via optical coherence tomography. They divided 89 male mice (20 to 25 g) into four groups based on the duration of sound exposure [19]. The results indicated an approximately 19.7-fold increase in the leakage of EB used as an albumin marker 1 h after sound exposure (*p* < 0.05) [19]. Leakage increased to 7.03 ± 0.10 μg/g of tissue from 0.37 ± 0.02 μg/g. In terms of the meningeal lymphatic vessels, 1 h after the opening of the BBB, the diameter of the lymphatic vessels increased 3.3-fold (3.00 ± 0.01 μm vs. 10.00 ± 0.07 μm, *p* < 0.05).

A similar study by Semyachkina-Glushkovskaya et al. [27] analyzed cerebral blood flow alterations resulting from sound-induced BBB opening using laser speckle contrast imaging. The study incorporated a subset population of 2-month-old mice in four groups (n = 10): before sound (control), 90 min after sound, 4 h after sound, and 24 h after sound. The BBB was opened via audible sound at 110 dB and 370 Hz in a 60-s on/off pattern for 2 h. FITC-dextran (70 kDa) was used in addition to EB to further characterize and analyze BBB permeability. In the 90-min period after sound exposure, there was significant leakage of EB into the brain, resulting in a 23.3-fold increase in permeability compared to the control group (9.10 ± 0.33 μg/g, *p* < 0.05).

Another study by Semyachkina-Glushkovskaya et al. [31] examined the relationship between stress-induced humoral and hemodynamic responses after sound-induced opening of the BBB associated with loud music. Indications of stress-related effects were measured through plasma levels of epinephrine. Analysis upon BBB opening 1 h after sound exposure showed that epinephrine decreased cerebral blood flow by 33 ± 5% (4.70 ± 0.50 A.U. vs. 7.00 ± 0.60 A.U., *p* < 0.001). The findings suggest that music exposure affects BBB permeability and an epinephrine-mediated response through vasorelaxation and decreased vascular tone, as seen in Table 1.

In a recent article by Semyachkina-Glushkovskaya et al. [29], efforts were made to examine and analyze the use of the noninvasive music-induced opening of the BBB associated with the clearance of fluorescent Aβ (Fαβ) using extended detrended fluctuation analysis (EDFA) through electroencephalographic (EEG) patterns. Eight groups were used, incorporating controls, EB, Fαβ, and FITCD groups and including groups used for EDFA for EEG pattern structure analysis. The study concluded that extravasation of OBB occurred 1 h after music exposure based on Eb in the brain parenchyma (2.84 ± 0.14 µg/g tissue vs. 0.12 ± 0.05 µg/g tissue, *p* < 0.001). In addition, music-exposed mice showed faster extravasation of Fαβ from portions of the brain than the intact BBB mice. The study also concluded that the EEG patterns could be an effective tool to measure/identify BBB integrity, as well as drainage functions of the brain.

### 4.2. Music and Neurodegenerative/Neurological Injuries

Music interventions alleviate the collateral effects of brain injuries, such as motor dysfunction and decreased social interaction. In addition, music has a developmental impact on neurodegenerative diseases such as Alzheimer disease. In a sample of people with traumatic brain injuries or stroke, recovery via music therapy was used as an alternative to basic therapy to assess whether social interactions improved [30], as summarized in Table 2. The study used human subjects and included 18 individuals (6 men and 12 women); 4 men and 6 women were included in the experimental group. Participants were assessed on a 7-point scale by family members and staff to determine whether their moods or behaviors improved. The results indicated that the music group had significantly more involvement in therapy and that these participants had improved mood (*p* < 0.01). Therefore, music positively affected the participants by elevating their moods and social interactions. Additionally, the participants were more likely to be involved in rehabilitation (*p* < 0.06).

In another study, as outlined in Table 1 above, different doses and durations of music were used to measure the recovery of motor function after stroke in Sprague-Dawley rats [26]. After the rats were subjected to middle cerebral artery occlusion, they were split into the following groups: 1 h per day of music, 12 h per day of music, 12 h per 2 weeks of music, and a control group exposed to no music. Each group had 10 rats weighing between 220 and 250 g. The study showed that the group exposed to 12 h of music per day had significant improvements in motor function and higher brain-derived neurotrophic factor (BDNF) and glial fibrillary acidic protein (GFAP) levels than other groups (*p* < 0.05). In addition, the group exposed to 12 h of music per day showed a significant decrease in the modified neurological severity score (mNSS; *p* < 0.05). Researchers emphasized three main points: (1) music therapy can improve motor function following a stroke; (2) music therapy can lead to BDNF accumulation in the motor cortex following a stroke; and (3) music therapy can promote neuronal repair and improve brain plasticity. The findings suggest that music can be an effective way to improve poststroke motor dysfunction.

### 4.3. Supplemental Practices Involving Glymphatic Manipulation

There are many other approaches to opening the BBB. One of the more common is FUS with microbubbles [18]. Ultrasound accompanied with “lipid-shelled microbubbles” has proved to be a successful technique in manipulating the permeability of blood vessels in the vein [18]. Alongside the delivery of microbubbles could be ultrasound contrast agents, which are drugs that are injected into the brain. FUS may be an effective method to increase the permeability of the blood–brain vasculature. The pores and intracellular gaps in the BBB endothelium are highly dependent on acoustic pressure [18]. The interventional outcomes of FUS can be visualized in Figure 1 because this technique also applies in terms of BBB opening and an increase in glymphatic function. In vitro experimentation led to this discovery, which has been used in areas such as skeletal muscles, the heart, and the BBB. It is also important to note that although FUS has been shown to promote positive outcomes in terms of drug delivery, there has been evidence indicating neuroinflammation due to FUS-BBB disruption [32]. This has opened the door to discovering the extensive physiological responses involved in BBB manipulation.

In addition to FUS, sleep has been associated with an increase in Aβ clearance and the sleep-wake cycle regulating glymphatic clearance, as presented in Figure 1. A 60% volume increase was seen in the cortical interstitial space during sleep, which resulted in efficient convective clearance of Aβ and other compounds. Furthermore, the volume increase was accompanied by activation of macromolecular diffusion in brain tissues. As a result, sleep facilitates the clearance of metabolites from the brain, which can help delay the onset of neurodegenerative diseases, such as Alzheimer-related vascular dementia.

There is evidence that music influences sleep quality, which may affect glymphatic clearance [26]. This study was composed of six mice evaluated using in vivo two-photon imaging. The results indicated that Aβ was cleared twice as fast in the sleeping mice as in the awake mice. In addition, analyses indicated more than a 60% increase in the cortical interstitial space during sleep, resulting in efficient convective clearance of Aβ and other compounds.

### 4.4. Music and Mood

Regarding populations with Alzheimer disease and related dementias, there is a growing body of evidence that music interventions positively affect mood. For example, a systematic review and meta-analysis concluded that long-term music therapy reduced anxiety in people living with Alzheimer disease, as reported in combined results from seven studies using different evaluation instruments of anxiety (combined standardized mean differences: −0.64; 95% CI: −1.05 to −0.24; Z = 3.13, *p* = 0.002) [33].

Cheung et al. [34] conducted a randomized controlled trial concluding that 1:1 active music therapy reduced agitation in contrast to standard care by using the Cohen Mansfield Agitation Inventory nursing form, which is a self-reported Likert scale used to report the frequency of agitated behaviors from 1 to 7 (1 = never; 7 = several times per hour). The mean difference in score reported from baseline to week 14 of treatment between music therapy (−3.51) and standard care (3.26) was −6.77 (95% CI: −12.71 to −0.83; *p* = 0.027) with a medium effect size (*d =* 0.50) in people living with dementia.

Bakerjian et al. [35] similarly reported improvements in mood and behavior in a prospective, mixed-methods cohort study with 4107 enrollees over a 3-year period in 265 nursing homes. These outcomes are presented in terms of psychosocial effects, as seen in Figure 2. They found a reduction in aggressive behavior (−0.20 ± 0.02, *p* < 0.001), as well as in the use of antipsychotic (−0.11 ± 0.03, *p* < 0.001), antianxiety (−0.17 ± 0.03, *p* < 0.001), and antidepressant (−0.09 ± 0.03, *p* < 0.001) medications. They concluded that a resident-level control group was necessary for follow-up work (the study did not include one).

### 4.5. Variabilities in Intensity and Exposure Time

Studies varied between intensity and exposure time to music, sound, or noise. Semyachkina-Glushkovskaya et al. [20] showed that music/sound exposure had different effects at 70 dB vs. 90 and 100 dB and at 1-h durations vs. 4- and 24-h durations. The lower stimulus had a similar effect on BBB permeability as the control group (no music/sound stimulus), whereas both 90- and 100-dB exposure increased BBB permeability. The permeability was assessed by the transfer of low- and high-weight molecules. Exposure for 1 h to music/sound at 90 and 100 dB affected BBB permeability to contrast after 4-h and 24-h exposures to a greater extent than at the same volume level.

Semyachkina-Glushkovskaya et al. [27] showed variable effects of music/sound intensity and exposure time on BBB permeability, as measured by changes in cerebral blood flow. The study assessed cerebral blood flow at 90 min, 2 h, 4 h, and 24 h after exposure to sound. Volume intensities of 110 dB and a 370-Hz pure tone stimulus significantly affected BBB opening, particularly after a 2-h exposure. For example, a 90-min exposure at 110 dB and 370 Hz led to a 23.3-fold increase in permeability compared to the control group. Semyachkina-Glushkovskaya et al. [31] showed variable effects on stress-induced humoral and hemodynamic responses after the sound-induced opening of the BBB associated with loud music. The results were assessed by measuring epinephrine activity. After a 1-h exposure at a loud volume (100 dB), measurement and analysis of epinephrine showed a decrease in cerebral blood flow.

### 4.6. Effects of Music and Sound on BDNF Production

BDNF is a neurotrophin that modulates the growth and function of neurons in the central nervous system and can thereby influence hypothalamic function [25]. Music has been seen to affect the ANS and can be measured by observing the change in hypothalamic function caused by BDNF. Mice exposed to 6 h of slow rhythmic music for 21 consecutive days showed elevated levels of BDNF compared to controls [25]. Listening to music induces neurochemical changes and involves cognitive components with distinct brain substrates [25]. It is important to note that there could be differences between the effects of harmonic music and other auditory stimuli, which is a significant limitation of this study [25].

Another study shed light on this limitation; however, it measured BDNF production differences in the rat womb when exposed to Mozart music composition in its default sequence and in a reverse sequence. There were significant differences between average BDNF production in the cerebral hemispheres when Mozart’s composition was played in default and in reverse (8.98 ± 1.31 ng/mL default sequence vs. 5.58 ± 3.08 ng/mL reverse sequence, *p* = 0.004) [36]. Furthermore, in another study by Marzban et al., newborn rats were exposed to slow-rhythm Mozart Sonata music. The BDNF concentration in the hippocampus of these rats was considerably higher than in control rats (*p* ± 0.01). The concentrations of BDNF in the control and music exposure groups were 86.30 ± 2.26 and 94.60 ± 6.22 ng/g wet weight, respectively [37]. This study concluded that exposing mice to Mozart music early in life can lead to higher BDNF concentrations in the hippocampus.

BDNF and its receptor, the tyrosine kinase receptor B (TrkB), aid the central nervous system by regulating synaptic plasticity by producing the TrkB protein, and overexpression of this protein has been shown to improve cognitive skills [38]. Because music is known to positively affect cognition [38], one can assume that music’s benefit on neuronal plasticity may come from the BDNF/TrkB pathway. This study aimed to show the biological and molecular mechanisms of the benefits of music exposure. It demonstrated that perinatal exposure to music enhanced learning performance in adult mice, with high TrkB protein levels in the cortex, mainly in the auditory areas [38].

## 5. Discussion

This review primarily evaluated music- and sound-related noninvasive techniques and the relationship between the activity of the BBB and the glymphatic system. The findings show a strong correlation between sound interventions and the activity of the MLS/glymphatic system, including the BBB. Based on existing literature, sound interventions have the ability to open the BBB and potentially allow clearance of toxins from inside to outside the brain. Furthermore, after the opening of the BBB, there is an increase in the meningeal vessel diameters, triggering meningeal lymphatic flow to the lymph nodes. The opening of the BBB is stimulated by several sound interventions, as shown in this review. Music interventions significantly improve mood and serve as a potential therapy among Alzheimer disease and dementia patients, which may be associated with the activity of the BBB and glymphatic system. It may be that accumulation of Aβ within the brain due to several neurological diseases such as Alzheimer disease can be further reduced with sound to enable clearance through the MLS/glymphatic pathways.

Until recently, music was rarely used as medicine in the clinical setting. Music has a wide-ranging potential to affect the neurological and cardiovascular systems; the clinical applications have yet to be fully explored. The current findings of the effects of music on the glymphatic system suggest that it can be a beneficial practice in the clinical setting. In addition, evidence has suggested that high-intensity music and sound could serve as an intriguing alternative in terms of a low-cost and noninvasive form of drug delivery to treat neurodegenerative diseases [28]. Despite the benefits that music/sound can have in improving clinical practice, much remains to be discovered regarding safety, side effects, and challenges associated with any intervention of the brain and heart.

Findings indicate that music is an effective tool to modulate brain and body functions and that music has a positive impact on mood, autonomic and cognitive function, and neuronal health. The review showed that there are still gaps in research because many of the articles are focused on ultrasound and its correlation to the opening of the BBB. Information on the impact that music and acoustic sounds have on the opening of the BBB is limited, showing that this is an area that needs to be further explored and evaluated. Development of validated and repeatable imaging protocols to better assess the effects of music interventions on functional changes in metabolic waste clearance via glymphatic flow is needed. A limited amount of information is available on the impact of music and sound on BBB opening, thus showing that this area needs further exploration, validation, and rigorous evaluation.

### 5.1. Limitations

This review included rats, mice, and humans and highlights music’s effect on their glymphatic system clearance. Therefore, to draw accurate correlations, it is essential to consider the anatomical and functional differences between rodents and humans regarding how they are affected by symphonies and harmonies. This raises the question of whether rats or mice should be used and how well they mimic the human body and behaviors.

Another limitation is the lack of variation in music among the studies. It is essential to note in future studies how different genres and music styles can affect individuals’ behavior and the permeability of the BBB. Comparing symphonies and hard rock music could help establish a better connection between music and its effect on behaviors. Due to the limited literature on the correlation between music and the glymphatic system, results from preclinical work should be used to address further the impact of the permeability of the BBB on waste clearance and its connection to the onset of neurological diseases in humans.

### 5.2. Future Directions

Music interventions can be applied to numerous health care areas, as demonstrated by the articles included in this review. Future studies in both humans and rodents can solidify this relationship and further evaluate the effect of rhythm, pitch, and frequency of music as an intervention for enhancing the glymphatic system to increase the clearance of waste from the brain. In addition, future studies can broaden the scope of this research to assess the connection between music and sleep on the glymphatic system. Music and sleep impact the clearance of waste, thereby reducing the occurrence or slowing the progression of neurological diseases, such as Alzheimer disease.

Potential areas of exploration include the development of experiments that will yield accurate and reproducible results. This raises questions about designing a double-blinded placebo trial to optimally address these issues. Studying clearance in the glymphatic pathway has been a challenge in humans. Advances in protocols have allowed investigators to understand the data better. Additional work should be done on various molecules that can affect the permeability of the BBB. This includes researching molecules to determine the impact on the opening of the BBB and waste clearance in the brain.

Regarding future research, the intensity and exposure time of sound/music are areas of high importance for understanding the optimal points to affect the brain, specifically the BBB and meningeal lymphatic/glymphatic system. Intensity and exposure time were not presented consistently enough in the current literature to allow for specific comparisons of their effects on BBB opening and lymphatic/glymphatic clearance. The choice of these variables was random; however, these data are important for future studies because they reflect direct interventions that produced the results analyzed in this review. Considering that intensity is measured in decibels and duration, this is an area that can be expanded in the future. Such work could lead to future rigorous studies promoting a more efficient categorization of sound/music and its impact on the brain.

Furthermore, BDNF seems to be involved in the activation pathways after music exposure; future studies should examine proteomics in different brain parts, different cell types of the brain, and potentially even in the CSF and serum. Studies should also assess effects at other times after music exposure. Such future preclinical and clinical studies should clarify the anatomical and cellular changes that are activated after increases in protein production and the regulation of clearance through the meningeal lymphatic/glymphatic system or through other pathways.

Additional studies can be done to address whether music has different effects based on sex or age. Experimental work can include diverse participants, providing insight into differences between men and women and facilitating comparisons between music and other behaviors, such as sleep and mood, on the glymphatic system in various age groups. Sleep is also a constant concern in older adults; all of these variables should be further investigated and integrated into future trials.

## Figures and Tables

**Figure 1 brainsci-12-00742-f001:**
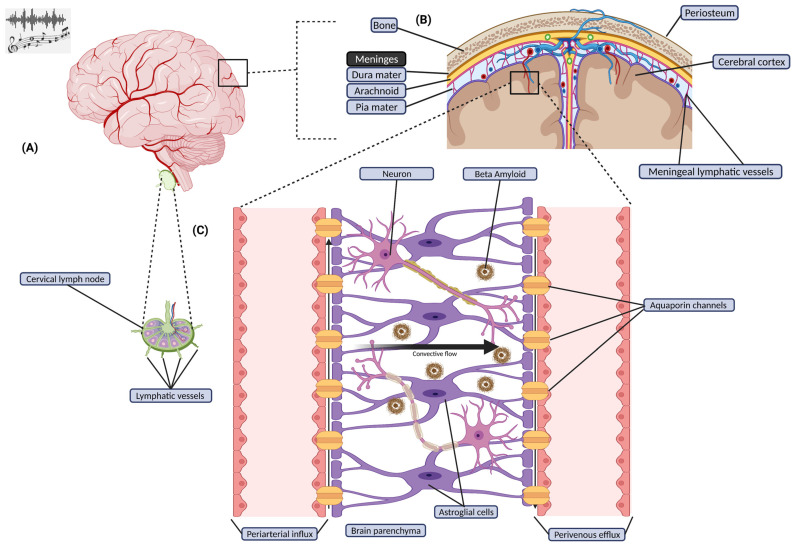
Glymphatic/Meningeal Lymphatic System Process. Illustration of the glymphatic/meningeal lymphatic system in terms of the cerebrospinal fluid (CSF) and the movement of solutes in and out of the brain. An expanded view is provided to allow for a better understanding of the process involved with the meningeal lymphatic vessels in reference to the blood–brain barrier and the flow of CSF. (**A**) Music/sound exposure in the scope of brain structure in terms of the CSF as paravascular influx is involved in the meningeal lymphatic system exchange from arteries to veins. A deep cervical lymph node representation is shown in which meningeal lymphatic vessels originate from within the neck. Nodes are involved with waste elimination based on lymph flow and perivenous efflux pathways. (**B**) A zoomed-in representation of brain meninges in which the meningeal lymphatic vessels are located in the subarachnoid space containing the CSF. A clear separation of meningeal lymphatic anatomy and its process is depicted compared to its counterpart, the glymphatic system residing in vessels in the brain tissue, as shown in panel **C**. (**C**) A zoomed-in representation of the glymphatic system based on the perivascular space in brain tissue. Brain circulation in terms of periarterial influx and perivenous efflux provides a pathway for solute waste clearance. The pia mater layer is tied with astrocytes and aquaporin-4 channels (AQP4). CSF influx through periarterial space into arteries allows AQP4 water pump function to drive CSF/interstitial fluid (ISF) exchange within the brain parenchyma. The convective flow of CSF/ISF exchange drives interstitial solutes through opposite AQP4 channels and through paravenous spaces into venous vessels as efflux essentially clears solutes/waste from the brain.

**Figure 2 brainsci-12-00742-f002:**
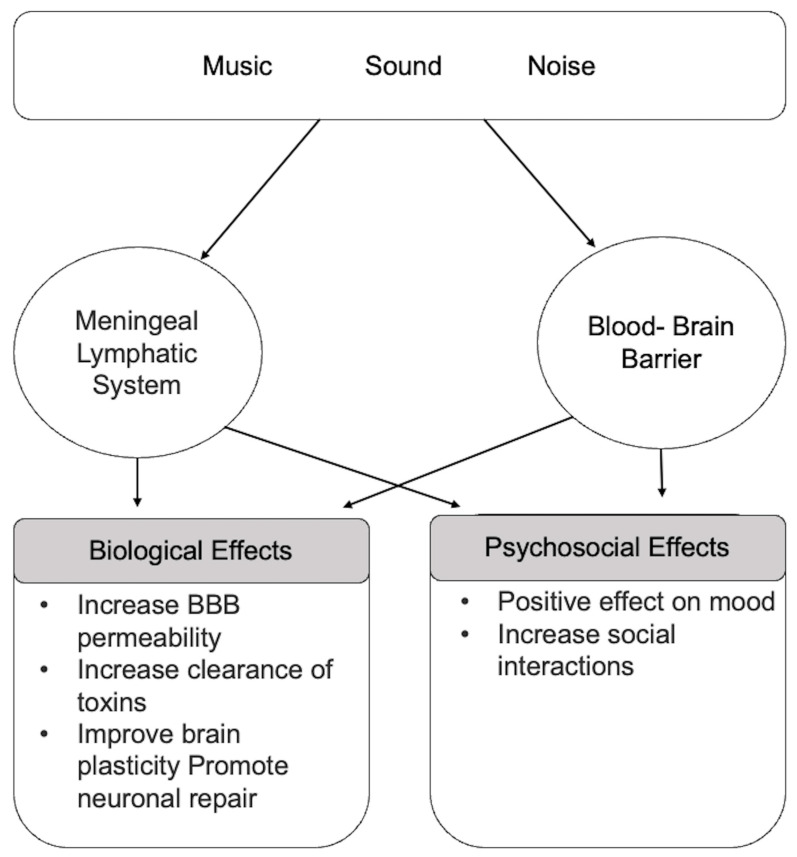
Relationship between music/sound exposure on meningeal lymphatic system/glymphatic system and blood–brain barrier function. The illustration depicts important outcomes that arise from music/sound exposure on the meningeal lymphatic/glymphatic system. The bullets emphasize the main points of these studies and this review.

**Table 1 brainsci-12-00742-t001:** Studies provide insights into music/sound and rodents’ meningeal lymphatic/glymphatic system. Overview of all studies that showcase the effect of music on the glymphatic system. Each study is broken down based on the overall purpose and the selected sample, design, and study outcomes.

Reference	Test Subjects	Research Methodology	Anatomical/Functional Outcomes
[25] Music exposure differentially alters the levels of brain-derived neurotrophic factor and nerve growth factor in the mouse hypothalamus.	Disease Condition: None Species: Mouse Strain: BALB/c (Charles River, Italy) Weight: 30 g Avg. Age: 40 day N: 20	Mice were exposed to slow rhythmic music for 6 h for 21 d. Music had mild sound pressure levels, around 50–60 dB. Control mice were placed in a similar environment but without music. Animals had free access to food and water and were placed on a 12 h light/dark schedule. The music was played during nighttime hours due to the mice being nocturnal. On day 22, mice were sacrificed, and the hypothalamus was extracted to measure BDNF and NGF production levels. Sound: Slow rhythm music (~50 and 60 dB)	Compared to the control mice, the music-exposed mice had significantly increased (32 vs. 55 ng/g, *p* < 0.01) BDNF production in the hypothalamus. Music also had a significant effect (49 vs. 30 ng/g, *p* < 0.05) on NGF production, but NGF levels conflicted with BDNF changes. In the music-exposed mice, NGF levels were less than those of the control mice.
[26] Music therapy alleviates motor dysfunction in rats with focal cerebral ischemia-reperfusion injury by regulating BDNF expression.	Disease Condition: Focal cerebral ischemia-reperfusion injury Species: Rat Strain: Sprague-Dawley (Nantong, China) Weight: 220–250 g N: 90 Sex: M	Rats underwent middle cerebral artery occlusion (MCAO) for 1h followed by reperfusion. Rats that survived were separated into 4 groups: MCAO 1 h/d 12 h/d Accelerated music group with music for 2 wks Nissl staining was performed on infarct zones and the motor cortex and immunofluorescence BDNF and GFAP. Sound: Mozart K.448 (65–75 dB)	In comparison to the MCAO and 1 h group, data indicated that the motor function in the 12 h/d music group was significantly enhanced. Music therapy notably lowered the focal neurological deficits in the 12 h music group 14 d and 21 d post-MCAO (*p* < 0.001). On days 14 and 21, the mNSS of the 12 h music group was less than the 1 h music group (*p* < 0.05). Immunofluorescence assay revealed BDNF accumulation in newborn and astrocyte cells in the music group, and GFAP fluorescence showed substantial intensity in the music group compared to the MCAO group.
[19] Application of optical coherence tomography for in vivo monitoring of the meningeal lymphatic vessel during the opening of blood–brain barrier: mechanisms of brain clearing.	Disease condition: None Species: Mouse Strain: N/D Weight: 25 g N: 89 Sex: M	Evans Blue and FITC-Dextran were used to monitor large and small molecule intravasation through the BBB, respectively. Analyses of Evans Blue were performed in 4 groups: before sound (control) 1, 4, 24 h after sound exposure FITCD extravasation was measured with imaging results only; no statistical differences were reported. First, the audible sound was administered for 60 s, followed by a 60 s pause for 2 h. Then 1 h after sound exposure, FITCD was administered and circulated for 20 min, and then the mouse anatomies were analyzed. Sound: Audible sound (100 dB, 370 Hz)	After 1 h of sound exposure, leakage of Evans Blue dye through the BBB had a 19.7-fold increase compared to the control group (7.30 ± 0.09 vs. 0.37 ± 0.02 μg/g of tissue, *p* < 0.05). In addition, after 1 h of sound exposure, the meningeal lymphatic vessels increased in diameter by (3.00 ± 0.01 vs. 10.00 ± 0.07 μm, *p* < 0.05). 4 h and 24 h after sound, BBB permeability recovered to normal conditions, EB extravasation decreased (0.54 ± 0.01 μg/g of tissue), and the BBB was impermeable to FITCD. FITCD extravasation through the BBB and accumulation in the perivascular space, including clearance, occurred within 30 min.
[27] Laser speckle imaging and wavelet analysis of cerebral blood flow associated with the opening of the blood-brain barrier by sound.	Disease Condition: None Species: Mouse Strain: N/D Weight: 20–25 g Avg. Age: 2 mo N: 40 Sex: M	Analyses of BBB permeability were conducted in 4 groups: no music 90 min after sound exposure 4 h after sound exposure 24 h after sound exposure Each group included 10 mice. To trigger the BBB opening, the audible sound was applied in 60 s on/60 s off intervals for 2 h. Then, EB was administered and circulated in the blood for 30 min; FITCD was administered and circulated for 2 min. Afterward, mice were sacrificed, and tissues were analyzed. Sound: Audible sound (110 dB, 370 Hz)	After 90 min of sound, leakage of EB in the BBB increased 23.3 fold compared to the control group (9.10 ± 0.33 vs. 0.39 ± 0.01 μg/g of tissue, *p* < 0.05). The measure of CBF increased by 51% for 90 min for cerebral veins compared with the control and only by 13% for microvessels compared with the control. Within 4 h of sound, BBB disruption was reversed to normal conditions. FITCD was used alongside Evans Blue as a characterization method to represent higher weight molecules; no statistical data were collected, just imaging. Results indicated sound at a certain degree of frequency and intensity could allow the BBB to open. EB and FITCD markers showed increased leakage and permeability into the BBB after 90 min upon sound exposure (*p* < 0.05).
[20] Loud music and specific sound stress open BBB.	Disease Condition: None Species: Mouse and Rat Strain: Mongrel and Wistar Weight: 20–25 g Avg. Age: “Wistar male rats of corresponding age been used.” N: 60 Sex: M	4 groups: control 1 h after sound exposure 4 h after sound exposure 24 h after sound exposure Music exposure for 2 h with 60 s on/off intervals ranging from 90–100 db. EB was used to measure the leakage of the BBB. The leakage of tracers was determined in 2 main groups (music and sound) divided into 4 subgroups. Music frequencies were 11–10,000 Hz, and the sound was 370 Hz. The focus was more on the effect of epinephrine and its impact on the BBB. Mongrel mice used are defined as wild mice with no known origin, thus known as “mongrel.” Sound: The music played was “Still Loving You” by Scorpions.	Data indicated music-/sound-induced an increase in BBB permeability based on EB, FITCD, and Gd-DTPA markers. After 1 h of sound exposure, leakage of EB increased 17.3 fold (music) and 18.6 fold (sound) (*p* < 0.001). Similar results for 90 db, and no change for 70 dB. Optimal time of BBB permeability 1 h after music/sound exposure (100 dB). Natural factors, including loud sounds and music, reversibly open the BBB for high-/low-weight molecules and 100-nm liposomes. However, the opening was short, followed by a quick recovery. Results relating to epinephrine showed an increase of CBF by 19 ± 2% (11.00 ± 0.60 vs. 9.20 ± 0.70 A.U., *p* < 0.05) in the control group (before sound exposure), This increase led to vasoconstriction and an increase of vascular tone.
[28] Phenomenon of music-induced opening of the blood–brain barrier in healthy mice.	Disease Condition: None Species: Mouse Strain: Mongrel Weight: 20–25 g N: 50 Sex: M	4 groups: control no music exposure, 1 h after music exposure 4 h & 24 h after music exposure Mice exposed to music for 2 h with 60 s on/off intervals ranging from 90–100 dB. Exposed to biomarkers Gd-DTPA, FITCD, and EB under in vivo and ex vivo imaging to measure extravasation of BBB based on music exposure. Mongrel mice used are defined as wild mice with no known origin, thus known as “mongrel.” Sound: The music played was “Still Loving You” by Scorpions. The control group of mice (n = 15) was not exposed to the song or music.	Data indicated that there was clear extravasation of the music-induced opening in the BBB in 11 brain regions. Fluorescent microscopy of Evans Blue showed a visible permeability to the marker in the cerebral microvessels. Statistical data concluded (0.80 ± 0.03 vs. 0.58 ± 0.01 A.U., *p* < 0.001) extravasation for the cerebral microvessels. FITCD also showed enlargement of lymphatic vessels upon opening of the BBB. Statistical results indicated enlarged lymphatic vessels of deep cervical lymph nodes (22.30 ± 1.50 vs. 37.30 ± 2.00 µm, *p* < 0.001). The loud sound caused up to a 3.1-fold increase in the plasma level of epinephrine (immediately after the music) vs. the normal state (9.00 ± 1.50 vs. 2.90 ± 0.70 ng/mL, *p* < 0.001).
[29] A novel method to stimulate lymphatic clearance of beta-amyloid from mouse brain using non-invasive music-induced opening of the blood-brain barrier with EEG markers.	Disease Condition: None Species: Mouse Strain: Mongrel Weight: 20–25 g N: 74 Sex: M	8 groups: control intact BBB mice with EB and Fαβ music-induced opening BBB mice for EB and FITCD mice with the opening of BBB and Fαβ mice with intact BBB and Fαβ Mice were exposed to music for 2 h with 60 s on/off intervals ranging from 0–130 dB. Exposed to biomarkers: Fαβ, FITCD, and EB under in vivo and ex vivo imaging to measure extravasation of BBB based on music exposure. EEG markers were also used to monitor extravasation. Mongrel mice used are defined as wild mice with no known origin, thus known as “mongrel.” Sound: The music played was “Still Loving You” by Scorpions.	Results show extravasation of OBB 1 h after music exposure based on EB in the brain parenchyma (2.84 ± 0.14 µg/g tissue vs. 0.12 ± 0.05 µg/g tissue, *p* < 0.001). Music exposed mice showed faster extravasation of Fαβ from portions of the brain compared to the intact BBB mice. Faβ was 3.5 fold higher based on the dorsal to ventral part of the brain. (0.11 ± 0.03 a.u. vs. 0.25 ± 0.03 a.u., *p* < 0.001) (0.42 ± 0.01 a.u. vs. 0.12 ± 0.02 a.u., *p* < 0.001) They concluded that the use of EEG markers allowed and has practical use in identifying the opening of BBB.

**Table 2 brainsci-12-00742-t002:** Studies that provide insight into music/sound and the meningeal/glymphatic system in humans. Overview of all studies that showcase the effect of music on the glymphatic system. Each study is broken down based on the overall purpose and the selected sample, design, and study outcomes.

Reference	Test Subjects	Research Methodology	Anatomical/Functional Outcomes
[30] Effect of music therapy on mood and social interaction among individuals with acute traumatic brain injury and stroke.	Disease Condition: TBI; Stroke Avg. Age: 59.89 yr N: 18 Sex: 33% M Duration: 10 wks	18 individuals with TBI or stroke were assigned to one of the following conditions: music therapy control Participants, families, and therapists rated their moods based on the Faces Scale. In addition, one or two supplemental music therapy activities were involved, such as playing instruments, singing, improvising, composing, listening, and performing. Participants assigned to the control group received all of the standard therapies that are components of the inpatient rehabilitation regimen. Sound: Simple pitched instruments, including percussion and melodic. Singing and listening were also incorporated. Different instruments were used based on each participant’s preference.	As a result of the study, they concluded that music therapy had a beneficial effect on the behavioral and social outcomes of the participants with stroke or TBI and showed trends in respect to mood. Furthermore, in the standard rehabilitation process, the effect allegedly facilitated participation. The difference amongst the two groups on the Faces Scale was F = 3.46 (*p* < 0.01).

## Data Availability

Not applicable.

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
