# Peer review of "Effects of Sound Interventions on the Permeability of the Blood–Brain Barrier and Meningeal Lymphatic Clearance"

_brainsci, 2022, doi:10.3390/brainsci12060742_

Round 1

Reviewer 1 Report

The submitted review paper is very well written and of high interest for neuroscientists and other specialists. The manuscript presents the beneficial effects of music on blood-brain barrier permeability and meningeal lymphatic clearance and is suitable for publication in Brain Sciences after a few minor revisions:

Figure 1 - Figure should be cited in the main text; panel C seems cropped, since one text box seems incomplete.

Lines 78-79 - please check if both occurrences of word "average" are necessary.

Lines 93-94 - it is unclear, at least for me, what authors mean by "heart rate  volume and rhythm". Is it the cardiac output, as the product of heart rate and stroke volume?

Figure 2 - should be cited in the main text.

Line 188 - could be useful to provide a reference for "Wong's description of constructing a metanarrative review".

Table 1 and Table 2 - tables could be inserted in the results section, and cited in the main text.

Table 1 - the table header could be changed in order to be easier to follow. For instance, "Disease condition/Type of animal/Species/Weight/Average Age/Number(N)/Sex" could be changed to "Test subjects" and "Design/Method/Control Condition(s)/Music/Sound type" could be replaced with "Research methodology" or something similar. The same could be done also for Table 2.

Lines 243-244- please revise the phrase.

Line 273 - the molar mass of FITCD could be written between parentheses.

Lines 329 - please explain "FUS" abbreviation (I guess it's focused ultrasound).

Line 402 - the authors state that "the review was limited mainly to primates". However, the literature reviewed by the authors included only experimental studies on rodents and humans. 

The paper could be even more interesting and eye-catching if the authors would include a third figure that summarizes the effects of music and sounds on the brain at a cellular/molecular level (such as the increase in BDNF production).

Author Response

Reviewer 1 Comments:

The submitted review paper is very well written and of high interest for neuroscientists and other specialists. The manuscript presents the beneficial effects of music on blood-brain barrier permeability and meningeal lymphatic clearance and is suitable for publication in Brain Sciences after a few minor revisions:

Reply: The authors would like to express their appreciation for the reviewer’s time in providing helpful comments that have improved the manuscript. We are confident that with these improvements, this manuscript is now ready for acceptance in Brain Sciences.

Figure 1 - Figure should be cited in the main text; panel C seems cropped, since one text box seems incomplete.

Reply: Sorry for this error; the figure has been updated accordingly.

Lines 78-79 - please check if both occurrences of word "average" are necessary.

Reply: The reviewer is right, and we have fixed that sentence.

Lines 93-94 - it is unclear, at least for me, what authors mean by "heart rate volume and rhythm". Is it the cardiac output, as the product of heart rate and stroke volume?

Reply: We have rephrased the sentence to better define these terms. We now state that the cardiac ANS associates with parasympathetic activity as shown by heart rate variability measured through variation in time intervals between successive heart beats.

Figure 2 - should be cited in the main text.

Reply: Yes, we have now included the Figure 2 in the main text.

Line 188 - could be useful to provide a reference for "Wong's description of constructing a metanarrative review".

Reply: The Wong reference was used lower in the paragraph, but we agree that it should have been used in the stated sentence. This has been corrected.

Table 1 and Table 2 - tables could be inserted in the results section and cited in the main text.

Reply: We have added the Tables in the Results section and cited the individual tables as well.

Table 1 - the table header could be changed in order to be easier to follow. For instance, "Disease condition/Type of animal/Species/Weight/Average Age/Number(N)/Sex" could be changed to "Test subjects" and "Design/Method/Control Condition(s)/Music/Sound type" could be replaced with "Research methodology" or something similar. The same could be done also for Table 2.

Reply: We have updated both Tables accordingly.

Lines 243-244- please revise the phrase.

Reply: The phrase has been revised: “The search parameters were narrowed to focus on music interventions and meningeal lymphatic/glymphatic system relationships, resulting in eight references included in this review.”

Line 273 - the molar mass of FITCD could be written between parentheses.

Reply: We have added the molar mass of FITCD.

Lines 329 - please explain "FUS" abbreviation (I guess it's focused ultrasound).

Reply: Yes, FUS abbreviation stands for focused ultrasound. We have added the definition to the paper.

Line 402 - the authors state that "the review was limited mainly to primates". However, the literature reviewed by the authors included only experimental studies on rodents and humans. 

Reply: Indeed, the review was not limited to primates, and all papers included either mice, rats and humans, and we have updated such statement.

The paper could be even more interesting and eye-catching if the authors would include a third figure that summarizes the effects of music and sounds on the brain at a cellular/molecular level (such as the increase in BDNF production).

Reply: Instead of adding a figure, we have added a separate and dedicated a paragraph in the Results section on BDNF. We have also expanded on the effects of music and sounds on the brain at a cellular/molecular level.

Reviewer 2 Report

In this manuscript, the authors reviewed the literature and discussed the evidence of relationships between music, BBB permeability, and meningeal lymphatic clearance. The authors presented an overview of the anatomy and physiology of the system and discussed the uses of music to modulate brain and body functions, highlighting music’s effects on mood, autonomic, cognitive, and neuronal function.

This manuscript is well written and covers most of the literature regarding the effect of music on BBB permeability and clearance. I only have some minor suggestions:

  • Figure 1 has been cropped and some part of the original plot is missing on the right.
  • It is not very clear what is the purpose of Figure 2 and how authors are trying to present the connections. I suggest removing this figure completely or recreating it with more information for the audience.
  • I do recommend presenting the papers also based on the intensity of the sound and having a paragraph analysing those results to understand the sweet spot for the intensity that has the most effect. It seems at the moment that the intensity is quite random, however, it should be one of the most important factors. 
  • Moreover, understanding the logic behind the exposure time to the music is quite important. It would be great if the authors could explain and categorise the rationale behind those decisions in the experimental study design.

Author Response

Reviewer 2 Comments:

In this manuscript, the authors reviewed the literature and discussed the evidence of relationships between music, BBB permeability, and meningeal lymphatic clearance. The authors presented an overview of the anatomy and physiology of the system and discussed the uses of music to modulate brain and body functions, highlighting music’s effects on mood, autonomic, cognitive, and neuronal function.

This manuscript is well written and covers most of the literature regarding the effect of music on BBB permeability and clearance. I only have some minor suggestions:

Reply: The authors would like to express their appreciation for the reviewer’s time in providing helpful comments that have improved the manuscript. We are confident that it is now ready for acceptance in Brain Sciences.

Figure 1 has been cropped and some part of the original plot is missing on the right.

Reply: As mentioned above, this has been addressed.

It is not very clear what is the purpose of Figure 2 and how authors are trying to present the connections. I suggest removing this figure completely or recreating it with more information for the audience.

Reply: To make the figure clearer, we have added these terms to the left: sound intervention methods, components of the meningeal lymphatic system, and various outcomes of sound intervention methods on the glymphatic system. We believe this figure is a good summary of the overall effects of sound interventions, notably on the blood-brain barrier and the meningeal lymphatic system.

I do recommend presenting the papers also based on the intensity of the sound and having a paragraph analyzing those results to understand the sweet spot for the intensity that has the most effect. It seems at the moment that the intensity is quite random, however, it should be one of the most important factors. 

Reply: This is a good point, although the basic information related to the sound intensity is not presented consistently enough in the papers we reviewed to allow a comparison. Consequently, we have added a subsection regarding this topic in the Results section and have included a paragraph in the Discussion section describing sound intensity such that future studies can be independently validated and developed. As stated by the reviewer, the mention of intensity is quite random, and we have highlighted in the Discussion that this is an important factor for future studies.

The issue is that intensity likely refers to decibels, and how long and consistent that decibel sound is maintained is not described in depth within the referenced papers. This can be included in the discussion to expand on further research such that the studies can be rigorously compared with one another. Moreover, understanding the logic behind the exposure time to the music is quite important. It would be great if the authors could explain and categorize the rationale behind those decisions in the experimental study design.

Reply: This point is somewhat similar to the issue brought up above. The exposure time (and decibel level) are not consistently mentioned in the studies we reviewed. Accordingly, we have also included a suggestion in the Discussion regarding the need to conduct future rigorous studies so that better categorizations can be performed.
